# *Achillea fragrantissima* (Forssk.) Sch.Bip instigates the *ROS/FADD/c-PARP* expression that triggers apoptosis in breast cancer cell (MCF-7)

Abdulrahman Alasmari[1,2]*

1 Department of Biology, Faculty of Science, University of Tabuk, Tabuk, Saudi Arabia, 2 Biodiversity Genomics Unit, Faculty of Science, University of Tabuk, Tabuk, Saudi Arabia

* ab.alasmari@ut.edu.sa

**Data Availability Statement:** All relevant data are within the manuscript and its Supporting Information files.

**Funding:** The author(s) received no specific funding for this work.

## Abstract

*Achillea fragrantissima* is a shrub plant that belongs to the *Asteraceae* family in Arabia and Egypt. It is used as folk medicine and is a good source of phenolic acids, flavonoids, and some active compounds. To investigate the anti-cancer effect of *A.fragrantissima* on breast cancer MCF-7 cells and find the critical mechanism involved in apoptosis. The toxicity and pharmacokinetic studies of ethanolic extract of *A.fragrantissima* was examined for anti-breast cancer properties. In turn, cytotoxicity and cell viability were achieved by the MTT method. Furthermore, the trypan blue exclusion and microscopy examination proved the presence of apoptotic cells. Again, fluorescent staining such as AO/EtBr, DCFH-DA, Rho-123, and Hoechst-33342 reveals the cellular cytoplasmic disciplines upon *A. fragrantissima* effect. Moreover, cellular functioning tests like wound healing, colony formation, and Trans-well invasion assay were demonstrated. In addition, the qRT-PCR technique authenticates the *A. fragrantissima* -induced apoptotic network genes (*Caspase-3*, *Caspase-8*, *Caspase-9*, *Cytochrome c*, *BCL-2*, *BID*, *BAX*, *PARP*, *PTEN*, *PI3K*, and *Akt*) expression were evaluated. Mainly, the Immunoblot technique proved the expressed level of apoptotic proteins such as *cleaved PARP*, *CYCS*, and *FADD*. This study confirmed that the *A. fragrantissima* exerts cytotoxicity at 20 μg/mL for 24 hrs in MCF-7 cells. Also, decreases cellular viability, producing apoptotic cells and damaged cellular surfaces with dead matter. Consequently, it creates ROS species accumulation, loss of mitochondrial membrane potential, and fragmentation of DNA in MCF-7 cells. Furthermore, it arrests cell migration, induces colony-forming ability loss, and suppresses cell invasion. In addition, *A. fragrantissima* significantly upregulates genes such as *caspase-3, 9, cytochrome c*, *BID*, *BAX*, and *PTEN* while down-regulating the *Pi3K/ Akt* signaling. Nonetheless, *A.fragrantissima* induced *cleaved PARP*, *CYCS*, and *FADD* proteins in MCF-7 cells to avail apoptosis.

**Competing interests:** The authors have declared that no competing interests exist.

## 1. Introduction

Cancer is a significant disease in the human population, defined as the uncontrolled growth of cells with high metabolic changes. Breast cancer starts with breast tissue; In female society, it occupies the first place and surpasses lung cancer with 2.3 million new cases recorded in 2020 [1]. As of January 2022, breast cancer remains one of the most prevalent cancers globally, particularly among women. The current status of breast cancer research, diagnosis, and treatment is characterized by ongoing advancements in understanding its molecular subtypes, risk factors, and personalized therapies. Overall, breast cancer incidence rates vary across different regions and populations. Developed countries typically report higher incidence rates compared to developing countries, attributed to factors such as lifestyle, reproductive patterns, and access to healthcare. However, efforts in awareness, early detection, and improved healthcare systems have contributed to earlier diagnosis and better outcomes in many regions.

Epidemiologically, breast cancer incidence tends to increase with age, with the majority of cases diagnosed in women over 50 years old. However, it can also affect younger women, albeit less frequently. Certain risk factors have been identified, including genetic predisposition (e.g., mutations in *BRCA1* and *BRCA2* genes), hormonal factors (e.g., early menarche, late menopause), reproductive history (e.g., nulliparity, late age at first childbirth), lifestyle factors (e.g., alcohol consumption, obesity), and environmental exposures.

Efforts in epidemiological research aim to identify and understand these risk factors better, allowing for targeted prevention strategies and interventions. Additionally, ongoing research into the molecular mechanisms underlying breast cancer development and progression informs the development of novel therapies, including targeted therapies and immunotherapies, which hold promise for improving survival rates and quality of life for patients. Overall, while breast cancer remains a significant public health concern, progress in research, early detection, and treatment continues to improve outcomes for individuals diagnosed with the disease. Continued efforts in prevention, screening, and access to care are crucial in reducing the burden of breast cancer worldwide.

Apoptosis can be defined as cellular death and termed "Programmed Cell Death," It may lead by the *Fas-associated death domain protein* (29 KDa). It is a cytoplasmic adaptor molecule capable of transferring the apoptosis signal transduction to the caspase cascade [2, 3] and overexpression of *FADD* induces cell apoptosis in lung cancer cells [4].

Cellular apoptosis is a complex process and takes place in numerous proteolytic enzymes in action. Cellular apoptosis depends on the upregulation of the apoptosis gene and the downregulation of anti-apoptotic genes. In other words, upregulation of tumor suppressors or significant expression of pro-apoptotic genes in the cells [5].The mitochondria play a regulatory role in the intrinsic apoptotic pathway combined with the anti-apoptotic and pro-apoptotic proteins. The alteration of these proteins helps the oligomerization between *Bax/ Bak* proteins to form a hole in the outer membrane of the mitochondria to release the cytochrome c molecule as a hallmark of apoptosis [5]. In addition, numerous cellular signalings such as *MAPK*, *Akt*, *ERK*, and *NF-κB* help the cellular survive under toxic or oxidative cellular stresses [6, 7]. Modern research is turned into traditional herbal medicine or phytochemicals from edible sources for targeting cancer cells, activating cell apoptosis via different targeting of cellular signaling [8].

*Achillea fragrantissima* is a shrub plant known as lavender cotton [9] belonging to the *achillea* genus classified under the *Asteraceae* family, and it was used as folk medicine in Middle Eastern populations, particularly in Saudi Arabia [10, 11]. This herbal plant is used against many illnesses, including nasal infections, vision problems, viral infections (smallpox), fatigue, intestinal obstruction, fever, diabetes, and dysmenorrhea [10, 12].However, the anti-cancer property of this herb is understudied, and we assumed that *A. fragrantissima* ethanolic

extraction could be an antagonistic factor for breast cancer. The ethanolic extract of *A.fragrantissima* includes bioactive compounds such as Phenolic Compounds (flavonoids and phenolic acids), Terpenoids (sesquiterpenes and triterpenes), Alkaloids, Essential Oils (camphor, borneol, and eucalyptol), Tannins, and Acetylenes.

Herein, this study adopted MCF-7 cells to evaluate the cytotoxicity, cell viability, cell migration, cell invasion, and genes involved in apoptosis under the ethanolic extract of *A. fragrantissima*. The obtained data from MTT analysis proved that *A. fragrantissima* causes cytotoxicity and decreases cell viability. Further, DCFH-DA, Rho-123, and Hoechst analysis confirmed the generation of intracellular ROS, loss in mitochondria potential, and fragmented DNA, respectively. Furthermore, wound healing and transwell assay proved the reduction of cell migration and invasion in MCF-7 cells. Finally, the qRT-PCR and immunoblot data manifest evidence of induced apoptosis in MCF-7 cells. Taken together, *A. fragrantissima* could induce apoptosis in breast cancer MCF-7 cells, and these findings may be used in future cancer therapeutic studies.

## 2. Materials and methods

### 2.1. Chemical

The DMEM medium, Fetal Bovine Serum, and antibiotics were purchased from Himedia, Mumbai (India). Dyes such as Acridine orange, Ethidium Bromide, DCFH-DA, Rho-123, and Hoechst (33342) are procured from the Thermo Fisher Scientific Company (USA). All antibodies were procured from cell signaling technologies, USA.

### 2.2. Plant Collection, Identification, and extract preparation

In this study, *A. fragrantissima* plant ethanol extract was used to detect the apoptotic nature on MCF-7 cells. The mature *A. fragrantissima* plant leaves were collected in Tabuk, Saudi Arabia, and the prepared herbarium was identified and authenticated by Dr. Jacob Thomas, Taxonomist, Department of Botany and Microbiology, Riyadh, Saudi Arabia. The plant leaves were dried under dark conditions for seven days and were ground as a fine powder. The soxhlet apparatus was used to isolate the ethanol extract using 100 grams of fine leaves powder. The collected liquid was lyophilized and stored under -80°C until further use.

### 2.3. Cell culture

The human breast cancer MCF-7 cells and human embryonic kidney normal HEK-293 cells were purchased from American Type of Cell Culture (ATCC), USA. Cells were grown under DMEM with 10% FBS (V/V) and 1% penicillin/ streptomycin (W/V). The subculturing of cells was maintained at 5% $CO_2$ and 95% humidified atmosphere at 37° C. The fresh cultured cells were used for further experiments.

### 2.4. MTT assay

To determine the $IC_{50}$ value of the ethanol extract of *A. fragrantissima*, the MTT method was adopted. In brief, $1 \times 10^4$ cells of MCF-7 and HEK-293 were seeded in 96 multi-well plates overnight. The next day, the old medium was aspirated with a new fresh medium with differenced concentrated *A. fragrantissima* extracts. Then, the system was allowed for the next 24 hrs. Further, at the end of incubation, 20 µl of MTT substrate (5 mg/ 1 mL of 1× PBS) was added to each well and allowed for 4 hours. Then, the purple formazan crystal was diluted by adding 200 µl of DMSO. Finally, the plate was subjected to a multi-well plate reader at 595 nm.

The cell viability graph was obtained by following the formula, % of cell viability = OD of the sample/ OD of the control [13].

## 2.5. Examination of apoptotic cells by trypan blue exclusion method

For the assessment of apoptotic cells, the study conducted the trypan blue exclusion assay. Briefly, MCF-7 cells were seeded ($1 \times 10^5$ cells) on six-well plates and treated with *A. fragrantissima* for 24 hrs. The next day, cells were trypsinized and pellet. The harvested cell was subjected to cell counting using trypan blue on the cell counting chamber. The apoptotic cells were counted, and the apoptotic graph was plotted.

## 2.6. Cell viability

To test the viability nature of cells under *A. fragrantissima*, we used the MTT method for calculating cell viability. Different concentrations (0, 20, and 40 µg) of *A.fragrantissima* and time intervals (0, 12, 24, and 48 hrs) were utilized to affirm cell viability (Mosmann, 1983).

## 2.7. Phase-contrast microscopic examination

The phase-contrast microscopic analysis was used to identify the cells' morphological disturbance. In turn, cells were treated with a respective concentration of *A. fragrantissima* for intimated hours. Finally, the plate was analyzed under Accu-Scope microscopy, Life Technology, USA. The photographs were documented using a 20× magnification field.

## 2.8. Live/ dead assay (AO/EtBr staining)

The impact of *A. fragrantissima* was checked by analyzing the Live/Dead material under fluorescence microscopy. Briefly, $1 \times 10^5$ MCF-7 cells were seeded on six-well plates and allowed to overnight for maturation. Further, cells were treated with *A.fragrantissima* for 24 hrs. After the treatment, cells were incubated with AO/EtBr dual staining solution for 10 min in a dark room. Then the plate was analyzed under fluorescence microscopy at the magnification field of 20× (Accu-Scope, EXI-310, USA), and images were documented.

## 2.9. Determination of reactive oxygen species

For the determination of intra-cellular generated ROS, the DCFH-DA dye method was performed. First, MCF-7 cells were seeded on the six-well plate ($1 \times 10^5$ cells). After the maturation of cells, the cells were incubated under the *A. fragrantissima* for 24 hrs based on the $IC_{50}$ value. At the end of treatment, the plate was carefully processed by adding DCFH-DA solution and allowed for 30 minutes under darkroom conditions. Further, a liquid layer was removed, and the plate was analyzed under a green filter using Accu-Scope, EXI-310, USA. Finally, the captured images were documented. For spectroflurometer analysis, treated cells were detached and stained with DCFH-DA for 30 min and cells were resuspended with 1×PBS and subjected to Spectroflurometer examination. The OD values were recorded.

## 2.10. Mitochondrial membrane potential (Rho-123)

For the detection of the healthy mitochondria ($\Delta\Psi$m), we moved to examine the mitochondrial membrane potential by the Rho-123 staining. Briefly, $1 \times 10^5$ MCF-7 cells were plated in a six-well plated and treated with or without *A. fragrantissima* for 24 hrs. At the end of the treatment, the cells were washed with 1× PBS and stained with Rho-123. Continuously, the plate was placed under dark conditions for 15 minutes. After that, the excess dye was removed, and

the plates were analyzed under green light. Fluorescence microscopy captured the images and documents (Accu-Scope, EXI-310, USA).

## 2.11. Nuclear fate (Hoechst-33342)

For the detection of fragmented DNA, we used Hoechst (33342) staining method. Briefly, the MCF-7 cells were seeded on six-well plates and placed overnight in an incubator. The next day, cells were treated with or without *A. fragrantissima* for 24 hrs. Further, the plate was processed for the next, incubated with Hoechst dye for 15 minutes under dark conditions. Afterward, the plate was examined under a blue filter with a 20× magnification field (Accu-Scope, EXI-310, USA).

## 2.12. Wound healing assay

For the analysis of cell migration properties, we performed a wound-healing assay. First, we cultured a good monolayer of MCF-7 cells on the six-well plates. Based on that, the monolayer was wounded by a sterile pipette tip (p-10); afterward, the respective concentration of *A. fragrantissima* was added. Finally, and started the image documentation time-dependent using Accu-Scope, EXI-310, USA.

## 2.13. Matrigel-coated transwell assay

To test cell invasion properties, the matrigel-coated Transwell migration insert chamber was used. Briefly, the matrigel (2mg/ mL, BD Biosciences) was coated on the upper part of the chamber. The MCF-7 cells were seeded on top of matrigel, and it was treated with *A. fragrantissima*. Meantime the treatment hour, 10% FBS was added bottom of the chamber. The invaded cells from the upper part of the chamber were fixed with methanol for 20 minutes. The fixed cells are stained with 0.1% crystal violet solution. The invaded cells are counted by light microscopy at a 20× magnification field [14].

## 2.14. Colony formation assay

The colony-forming ability played a vital role in cell migration ability of cancer cells. Considering this, we performed colony formation by crystal violet staining. After exposure to *A. fragrantissima*; cells were collected and re-cultured for 15 days without any disturbance. Further, those colonies are stained with 0.1% crystal violet staining and counted. Finally, the graph was plotted.

## 2.15. qRT-PCR analysis

The qRT-PCR analysis determined the apoptotic network gene expressions. The *A. fragrantissima*-treated and non-treated cells were used to isolate the total RNA using the TRIZOL method. The 2µg of total RNA was used to construct the cDNA (cDNA synthesis kit, TAKARA, 6110A), and the syber green solution was used to find the specific gene expression (Used primers listed in Table 1). The β-Actin levels were utilized as the internal control. The relative quantitative gene expression was calculated by the $2^{-\Delta\Delta ct}$ method.

## 2.16. Immunoblot analysis

In the determination of specific protein expression, this study adopted immunoblot analysis. Briefly, cells were treated with or without an $IC_{50}$ value of *A. fragrantissima* and the whole cellular protein were isolated by the RIPA buffer method. Then, the protein was quantified using Lowry's method [15]. Finally, an equal concentration of protein was loaded on SDS-PAGE gel for protein separation. Then the gel was subjected to the protein transfer method using the

**Table 1. Primers were used in this study.**

| S. No | Gene | Forward primer | Reverse Primer |
|---|---|---|---|
| 1 | Actin-B | F:5'-TGAAGGCTTTTGGTCTCCCTG -3' | R:5'-ACAAAGTCACACTTGGCCTCA -3' |
| 2 | Caspase-3 | F:5'-TCCTAGCGGATGGGTGCTAT -3' | R:5'-CTCACGGCCTGGGATTTCAA -3' |
| 3 | Caspase-8 | F:5'-TTCAGACTGAGCTTCCTGCC -3' | R:5'-GACCAACTCAAGGGCTCAGG -3' |
| 4 | Caspase-9 | F:5'-AGGCCCCATATGATCGAGGA -3' | R:5'-GGCCTGTGTCCTCTAAGCAG -3' |
| 5 | BCL2 | F:5'-F-AAAAATACAACATCACAGAGGAAGT -3' | R:5'-GTTTCCCCCTTGGCATGAGA -3' |
| 6 | Bax | F:5'-GGAGCAGCCCAGAGGC -3' | R:5'-TTCTTGGTGGACGCATCCTG -3' |
| 7 | BID | F:5'-TGGGAGACGCTGCCTCG -3' | R:5'-GGAACCGTTGTTGACCTCAC -3' |
| 8 | CYCS | F:5'-TCGTTGTGCCAGCGACTAAA -3' | R:5'-ACCATGGAGATTTGGCCCAG -3' |
| 9 | PTEN | F:5'- AGGGACGAACTGGTGTAATGA -3' | R:5'- GGGAATAGTTACTCCCTTTTTGTCT -3' |
| 10 | PI3K | F:5'-CCCGATGCGGTTAGAGCC -3' | R:5'-TGATGGTCGTGGAGGCATTG -3' |
| 11 | Akt | F:5'-CAGGATGTGGACCAACGTGA -3' | R:5'-AAGGTGCGTTCGATGACAGT -3' |
| 12 | PARP | F:5'-CGCCTGTCCAAGAAGATGGT -3 | R:5'-CTGACTCGCACTGTACTCGG -3' |

nitrocellulose membrane. After that, the membrane was incubated with a specific primary antibody overnight. Then, the membrane was incubated with a secondary antibody for 4 hours. Finally, after a gentle wash, the membrane was allowed to react with BCIP/NBT substrate for a minute. A blots scanner documented the developed band.

## 2.17. Statistical analysis

All obtained results were presented as mean ± SD. p-value was considered statistically significant when $p < 0.05$, ****$p < 0.0001$, ***$p < 0.001$, **$p < 0.01$, *$p < 0.05$ and ns. One-way and Two-ANOVA were performed to find the significance between the groups.

# 3. Results

## 3.1. *Achillea fragrantissima* induces cytotoxicity in MCF-7 and HEK-293 cells

This study adopted the breast cancer cell MCF-7 and normal HEK-293 cells to explore the cytotoxicity nature of *A. fragrantissima*. The 24 hrs experimental times revealed that the *A. fragrantissima* induces the 50% cell death at 20 µg/ mL in MCF-7 cell. Moreover, the increasing concentration of extracts significantly induces cytotoxicity in MCF-7 cells, as shown in **Fig 1A**. Further, HEK-293 cells exhibited the IC$_{50}$ value at 30 µg/ mL at 24 hrs **Fig 1B**. The obtained data from the MTT method depicts that the *A. fragrantissima* is more toxic to breast cancer cells (MCF-7) than normal cells, and MCF-7 cells are subjected to further experiments in this study.

## 3.2. *Achillea fragrantissima* produces apoptotic cells

The apoptotic nature of MCF-7 cells was studied under the administration of *A.fragrantissima* through trypan blue exclusion method. The trypan blue exclusion assay data states that the concentration of *A.fragrantissima* increases; it induces apoptotic cells in a time-dependent manner. The 24 and 48 hrs death curves were higher than 0 hrs. The obtained data judged that increasing time produces apoptosis in MCF-7 under *A.fragrantissima* administration (**Fig 1C**).

## 3.3. *Achillea fragrantissima* decreases cell viability in MCF-7 cells

The cell viability was checked by the MTT method through different experimental times. The results revealed that the selected concentration of *A.fragrantissima* significantly reduced the

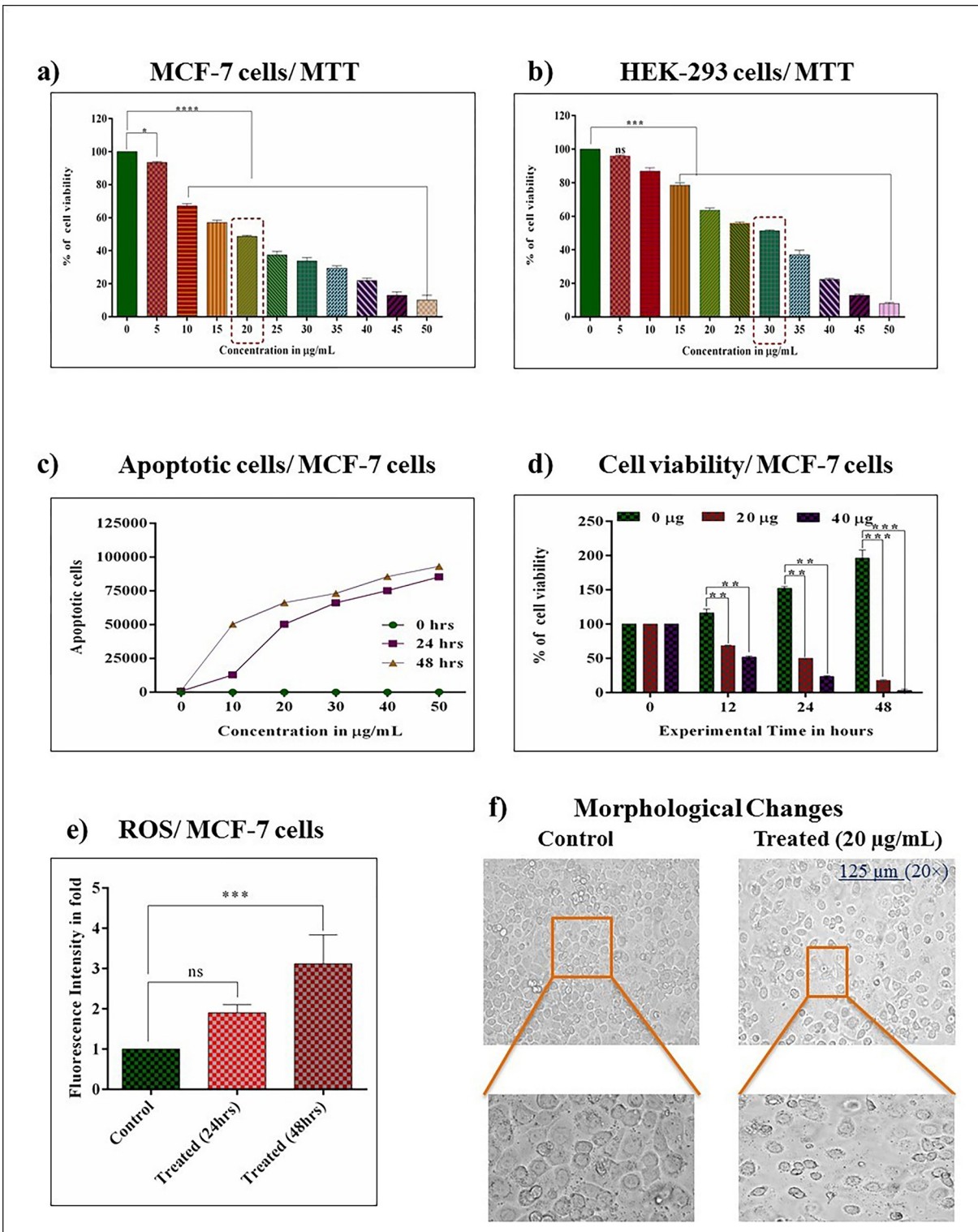

**Fig 1. Effect of *achillea fragrantissima* on cell cytotoxicity, cell apoptosis, cell viability, ROS generation, and morphological changes in breast cancer. a and b)** Show that *achillea fragrantissima* induced cell cytotoxicity in breast cancer (MCF-7) and normal cells (HEK-293), respectively. **c)** Shows the production of dead cells beneath *achillea fragrantissima* administrated. **d)** The graph illustrates the *achillea fragrantissima*-induced decrease of cell viability in MCF-7 cells at different time intervals. **e)** Shows the spectrofluorimeter analysis of the intracellular generation of ROS in MCF-7 cells. **f)** The images represent the *achillea fragrantissima* caused cellular structural damages in MCF-7

cells with differenced magnification. Values denoted mean ± SD for three independent performings (n = 3). Values are statistically significant at
\*\*\* $P<0.0001$, \*\* $P<0.01$, and ns. (0.05 Two Way ANOVA/ Turkey test).

viability power of MCF-7 cells by increasing experimental time and concentrations. Altogether, data suggested that the administration of *A.fragrantissima* is toxic to the MCF-7 cells, as shown in **Fig 1D**.

### 3.4. *Achillea fragrantissima* generates intracellular ROS

Cellular signaling is regulated mainly by the amount of produced ROS within the cellular cytoplasm of cells. Hence, we analyzed the intracellular ROS with a spectrofluorimeter. The results confirm that the *A.fragrantissima* triggers the intra-cellular ROS species in MCF-7 cells while increasing experimental time, as shown in **Fig 1E**. The obtained result clearly shows the *A.fragrantissima* induces the ROS accumulation in the cytoplasm of MCF-7 cells as result it may involve in the upregulation or downregulation of cellular function.

### 3.5. *Achillea fragrantissima* disturbs the morphology of MCF-7 monolayer

Morphology assessments of MCF-7 cells were documented after being treated with *A.fragrantissima*. The obtained images enforced that the control cells reflect good morphology and an increased number of cells. Meanwhile, the treated cells impair cell propagation and rupture the cell surface. The microscopic examination confirms that the *A.fragrantissima* more vulnerable effect on the MCF-7 morphology (**Fig 1F**).

### 3.6. *Achillea fragrantissima* instigates dead cells and ROS in MCF-7 cells

To enumerate cellular disciplines such as live/dead cells and intracellular ROS, we performed the AO/EtBr and DCFH-DA analysis by fluorescence microscopy. Briefly, **Fig 2 AO/EtBr, i)** showed excellent morphology and compact nuclei with significant emission of green light rather than red light. **Fig 2 AO/EtBr, ii)** reflects the weekend morphology with disturbed nuclei. Green and red color emissions were observed during the morphology examination. Moreover, DCFH-DA analysis depicts that the *A.fragrantissima* induces a higher level of ROS species in the treated MCF-7 when compared with the control MCF-7 cells. The control cells emitted a low level of green light **Fig 2 DCFH-DA, i and ii)**. The obtained data shows the *A. fragrantissima* can induce ROS species to produce dead cells in the MCF-7 cultures.

### 3.7. *Achillea fragrantissima* impacts mitochondrial health and DNA fragmentation in MCF-7 cells

The mitochondrial membrane potential plays an essential role in cellular energy production in cells. Keeping this in mind, we performed the Rho-123 staining to understand the effect of *A. fragrantissima* on mitochondria. The analyzed results explore the control cells emitting a significantly higher green emission than the treated MCF-7 cells. We assume that *A.fragrantissima* reduced the mitochondria' health and paved the way for energy depletion in the *A. fragrantissima*-treated MCF-7 cells (**Fig 2 Rho-123, i and ii**). Next, DNA integrity was cross-checked by the Hoechst-33342 staining method. Control MCF-7 cells do not emit blue light significantly, as they are compact nuclei and undamaged parts of the cells; control cells nucleus are does not loss their potential of neuclar envelop. In the case of treated cells, it emits blue light gratefully. By the observation, we judged that *A.fragrantissima* accelerated the DNA fragmentation in MCF-7 cells (**Fig 2 Hoechst-33342, i and ii**).

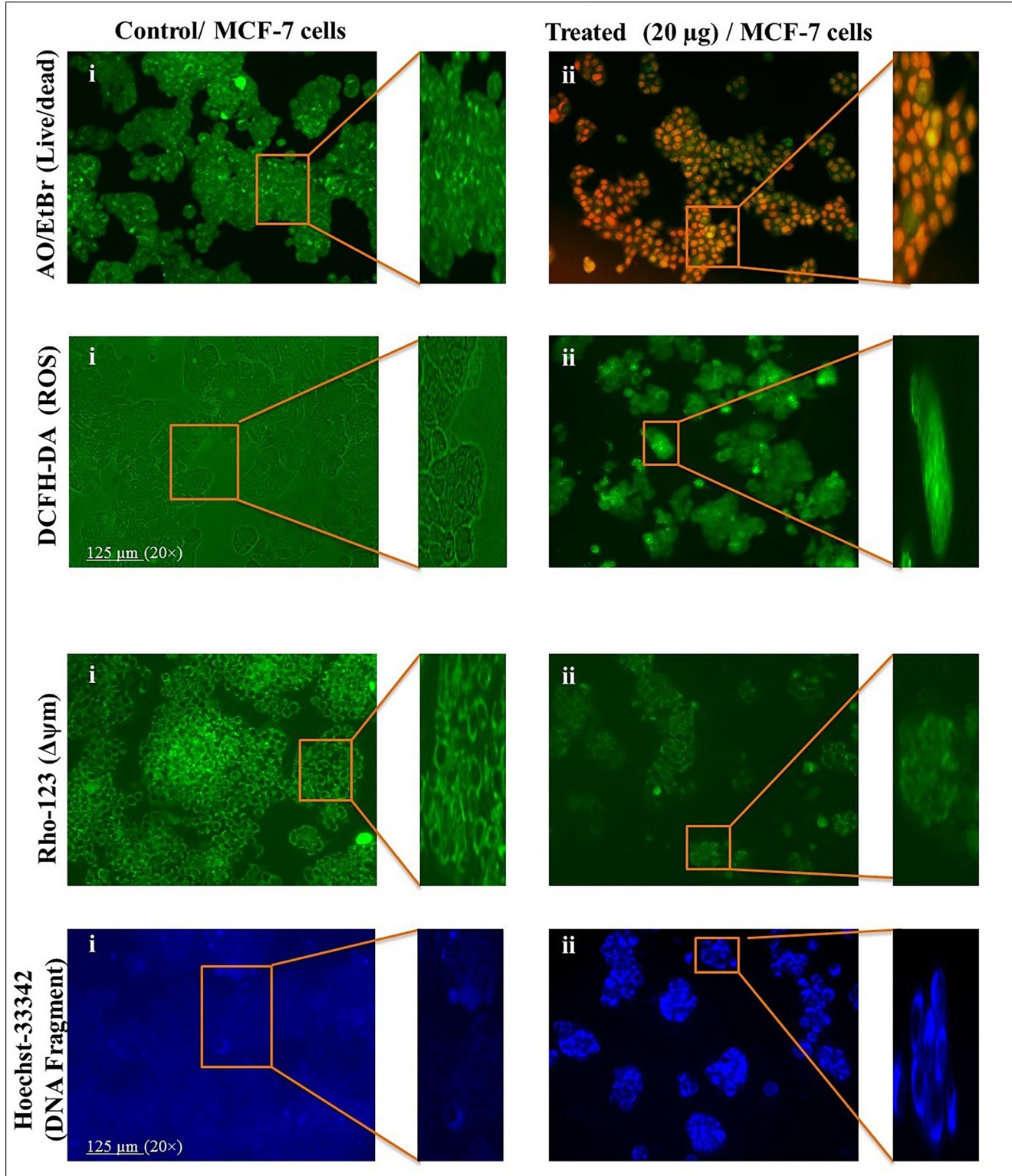

**Fig 2.** ***Achillea fragrantissima*** **triggers rupture in cell morphology (AO/EtBr), ROS generation, loss of mitochondrial potential, and DNA fragmentation in MCF-7 cells. AO/EtBr, i and ii)** Images reveal the MCF-7 morphological changes by the dual staining (AO/EtBr) method. Control cells exhibit green emission along with compact nuclei as well as good morphology of cytoplasm. On the contrary, *achillea fragrantissima*-treated cells exhibit red emission, shrinkage, loss of cell membrane potential, damaged nuclei, and monolayer cellular destruction. **ROS, i and ii)** Explain the higher intracellular generation of ROS in *achillea fragrantissima*-treated MCF-7 cells when compared to non-treated cells. **Rho-123, i and ii)** The Figure illustrates the

mitochondrial membrane potential of MCF-7 after the *achillea fragrantissima* treatment. The *achillea fragrantissima*-treated cells exhibit the loss of mitochondrial membrane potential when considered to control MCF-7 cells. **Hoechst, i, and ii)** The images represent the nuclear fate of MCF-7 cells under *achillea fragrantissima* treatment. Treated cells are accounted for the loss of nuclear membrane potential and disruption in nuclear integrity with enormous emission of blue emission. In the case of control cells, mild blue emission was documented. All images were snapped by 20× magnification using 125 μm.

### 3.8. *Achillea fragrantissima* inhibits cell migration in MCF-7 cells

To investigate the effect of *A.fragrantissima* on cellular migrationwound healing assay was performed in MCF-7 cells *in vitro*. As shown in **Fig 3A and 3B**, wound healing properties were diminished in the treated group. But control cells migrated actively and covered the wounded area rapidly. During the increasing interval, cellular integrity was lost in the treated group. Total findings illustrated the loss in cellular migration in MCF-7 cells induced by *A.fragrantissima*. The quantification data also reflects the suppression of the cellular migration ratio in MCF-7 cells.

### 3.9. *Achillea fragrantissima* impacts on colony formation of MCF-7 cells

Tumor re-births or tumor relapse was well-associated with the production of new colonies by the cancer cells. Based on that, the study conducted the colony formation assay in MCF-7 cells in *in-vitro* conditions. The results revealed that control cells have more capacity to produce large colonies, while decreased colonies were observed in the treated MCF-7 cells. The results account for the *A.fragrantissima* has more toxicity to the ability of colony formation in MCF-7 cells (**Fig 3C**).

### 3.10. *Achillea fragrantissima* suppresses the cell invasion (MCF-7) *in vitro*

Next, the invasion property of MCF-7 cells was examined by the Transwell chamber-matrigel coated assay. It was noted that active invasion was observed in the control MCF-7 cells and *A. fragrantissima* significantly reduced the active invasion among the treated MCF-7 cells. The finding concluded that the invasion ability was suppressed by the *A.fragrantissima* administration (**Fig 3D and 3E**).

### 3.11. *Achillea fragrantissima* impacts on *Caspase-3* expression

*Caspase-3* plays a significant role in "programmed cell death" among the caspase cascade family. Hence, the study needs to check the expression level of *caspase-3* mRNA under *A.fragrantissima* condition. The selected experimental time reveals a gradually increasing level of *caspase-3* mRNA in MCF-7 cells. The real-time PCR results coincide with the increased mRNA production in 4, 8, and 12 hrs (**Fig 4**). Conversely, immunoblot results reveal the null expression of caspase-3 (Data not shown) due to its genetics of MCF-7 cells [16].

### 3.12. *Achillea fragrantissima significantly* regulates anti, pro, and apoptotic genes in MCF-7 cells

It is a well-known key point that inducing cell death in the tumor would be a survival possibility among cancer patients. Thus, the expression levels of *Caspase-3*, *Caspase-8*, *Caspase-9*, *Cytochrome c*, *BCL-2*, *BID*, *BAX*, *PARP*, *E-Cadherin*, *PTEN*, *PI3K*, and *Akt*, were analyzed by the qRT-PCR which regulates apoptosis. After the treatment (24 and 48 hrs) with *A.fragrantissima*, the mentioned genes were analyzed using MCF-7 genomes.

As expected, apoptotic genes such as *caspase-3* and *9* mRNAs were expressed under the 24 hrs treated condition, and *caspase-8* mRNA expression was downregulated. On the other hand, the anti-apoptotic *BCL-2* mRNA level was significantly suppressed. Meantime, the pro-

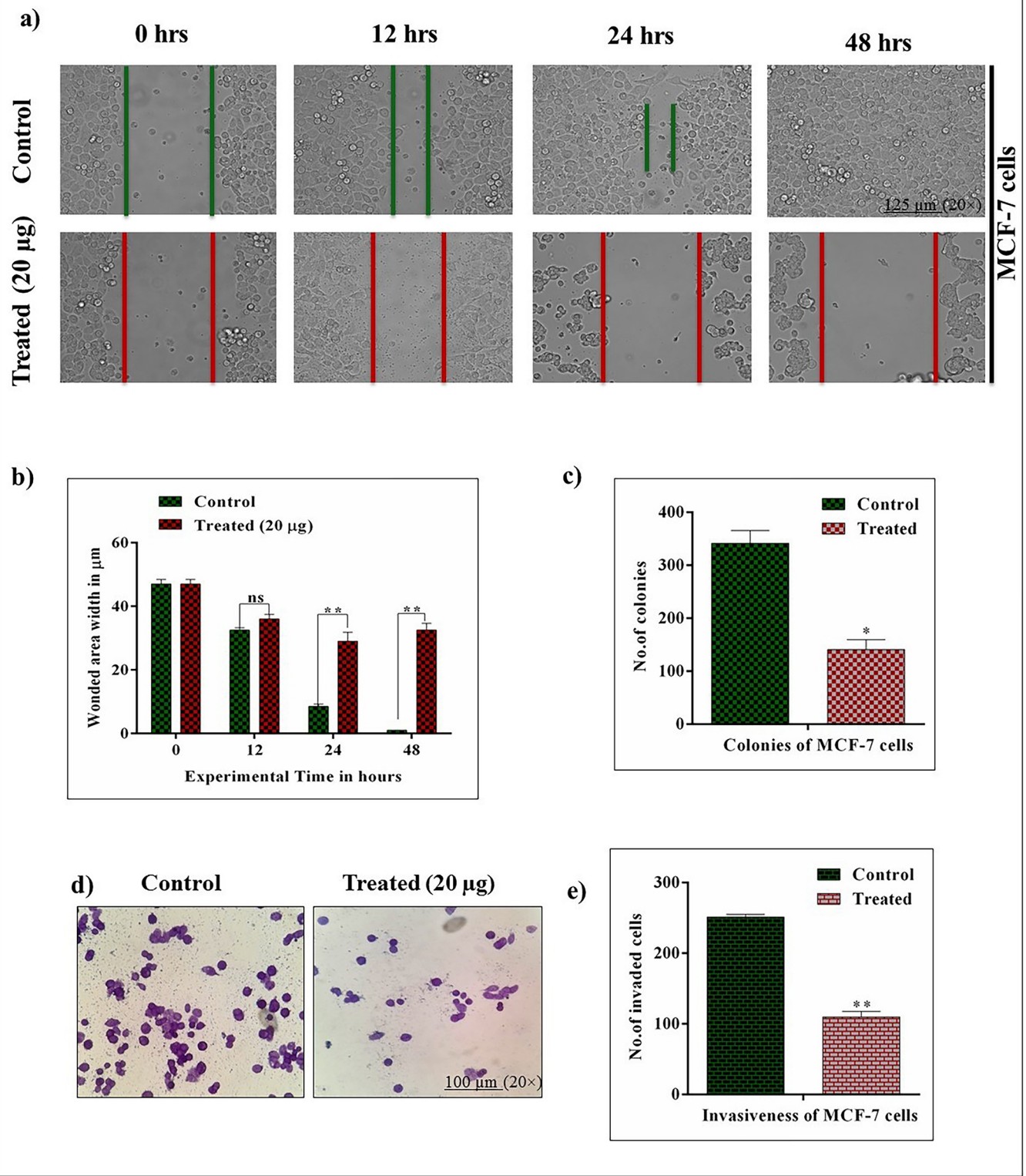

**Fig 3. *Achillea fragrantissima* actively inhibits migration, colony formation, and invasion in MCF-7 cells. a and b)** The impact of *achillea fragrantissima* on cell migration was analyzed by wound-healing assay. MCF-7 cells were exposed to the *achillea fragrantissima*; at the same time monolayer of the cell was gently scratched, and the distance of scratch was measured at 0, 12, 24, and 48 hrs by microscopy (125 μm and 20× magnification field). **c)** Shows the colony formation of the ability of MCF-7 cells in *achillea fragrantissima*-treated condition. **d and e)** The graph illustrate the Matrigel-coated invasion properties of MCF-7 cells. Control cells actively show invasion, and *achillea fragrantissima* -treated cells show the lesser invaded cells. Values are met significant at [#] $P<0.0001$, [***] $P<0.001$, [**] $P<0.01$, ns. ($p<0.05$ Two Way-ANOVA/ Dunnett's test).

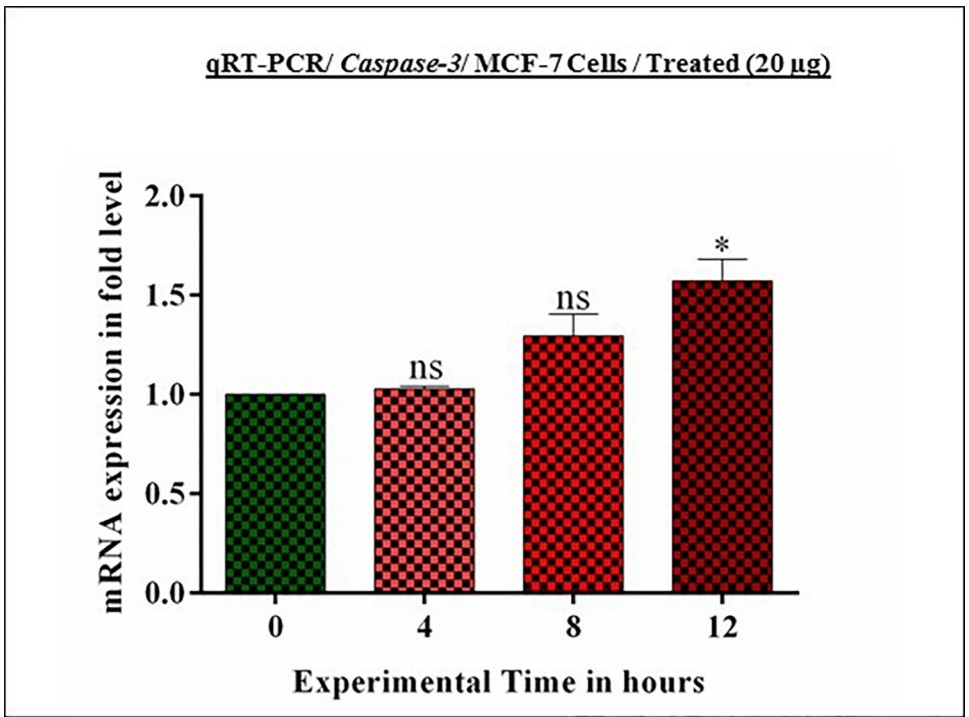

**Fig 4. *Achillea fragrantissima* upregulated *Caspase-3* mRNA in MCF-7 cells.** The qRT-PCR analysis revealed that the administration of *achillea fragrantissima* in breast cancer cells (MCF-7) stimulates the production of *Caspase-3* mRNA at different hours. Values denoted mean ± SD for three independent performings (n = 3). Values are statistically significant at [**] $P<0.01$ and ns. (0.05 Two Way ANOVA/ Dunnett's test).

apoptotic gene *BID* and *Bax* significantly elevated and acted as the initiator for apoptosis in MCF-7 cells. Moreover, *CYCS* (*Cytochrome c*) is crucial for regulating the ETC and thus releasing denotes cellular apoptosis. Hence, we analyzed the *CYCS* mRNA expression. Interestingly, the *A.fragrantissima* induces the expression of *CYCS* mRNA in MCF-7 cells. Furthermore, the study examined the *PARP* mRNA expression under the *A.fragrantissima* revealed that *PARP* expression was non-significantly downregulated in MCF-7 cells. In addition, *E-cadherin* mRNA level was downregulated under-treated conditions. Finally, a tumor suppressor *PTEN* mRNA expression was analyzed and significantly upregulated. In contrast, the expressions of *PI3K/Akt* mRNAs were suppressed (Fig 5). Fig 6 authenticated the genes expressions patterns as mentioned above; more considerably the same under the 48 hrs treatment. The obtained results revealed that the *A.fragrantissima* possibly regulates the apoptotic network genes at the mRNA levels in MCF-7 cells.

### 3.13. *Achillea fragrantissima* triggers *PARP, CYCS, Caspase-8*, and *FADD* proteins in MCF-7 cells

Cellular apoptosis is characterized by the upregulation of proteins such as *caspases*, *PARP* cleavage, and *CYCS* releasing. In other words, down-regulation of anti-apoptotic proteins along with stimulation of tumor suppressor protein. Keeping this in mind, we speculated that *A.fragrantissima* might upregulate *PARP, FADD, CYCS*, and *caspase-8* proteins in MCF-7 cells. In this regard, this study performed immunoblotting for specific protein detections. The blotting results confirmed that increasing time-interval produces *PARP*

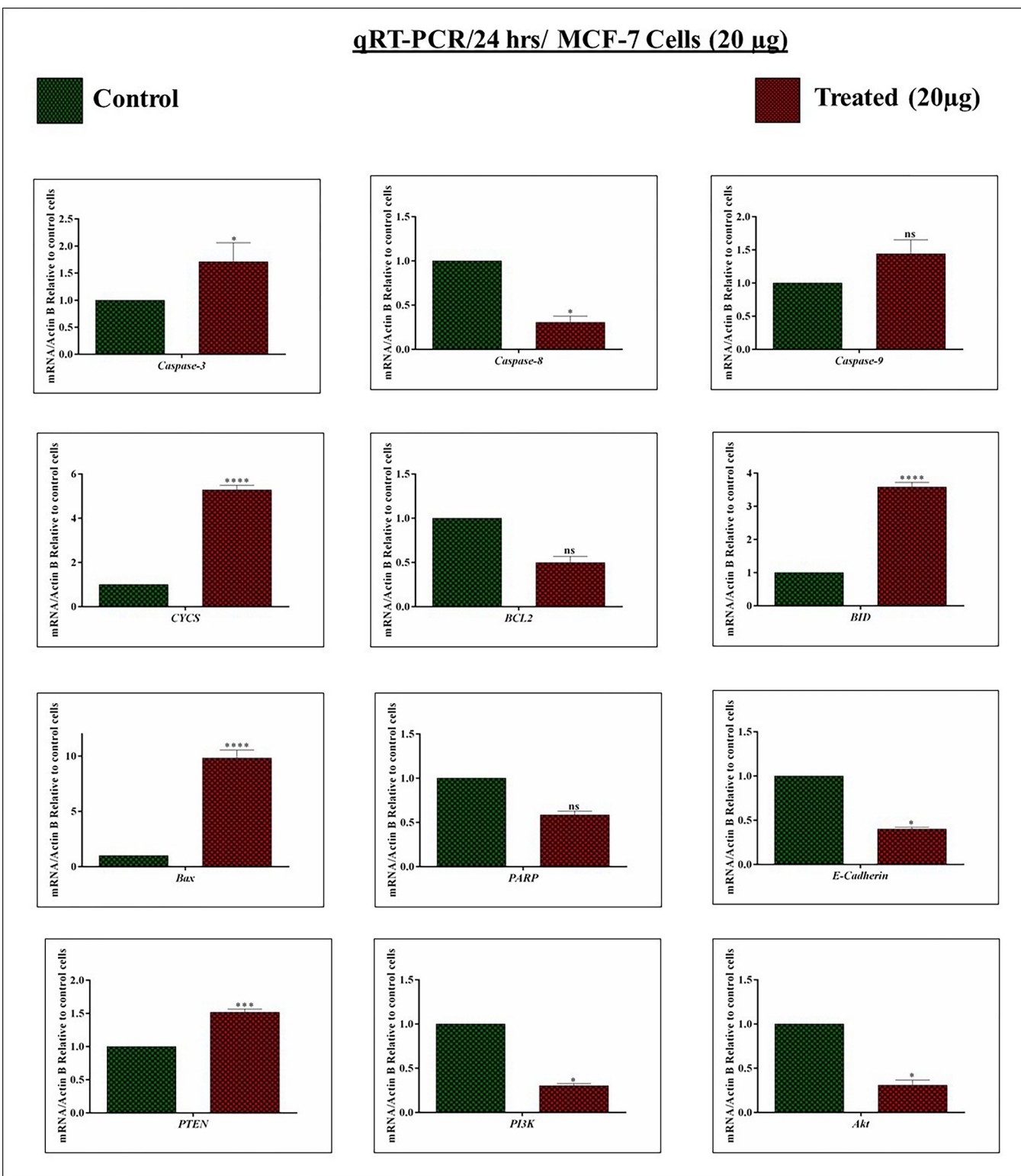

**Fig 5. Impact of *achillea fragrantissima* on apoptosis regulatory genes at 24 hrs (An mRNA account).** The MCF-7 cells were incubated with *achillea fragrantissima* for 24 hrs. The target mRNA such as *Caspase-3*, *Caspase-8*, *Caspase-9*, *Cytochrome c*, *BCL-2*, *BID*, *BAX*, *PARP*, *E-Cadherin*, *PTEN*, *PI3K*, and *Akt* expression was analyzed by qRT-PCR. The *β-Actin* level was utilized as an internal control. All values are represented as mean ± SD (n = 3). Statistically significant at [#] $P < 0.0001$, *** $P < 0.001$, ** $P < 0.01$, * $P < 0.1$ and ns. Statistical significance was performed by Two Way ANOVA/ Dunnett's test.

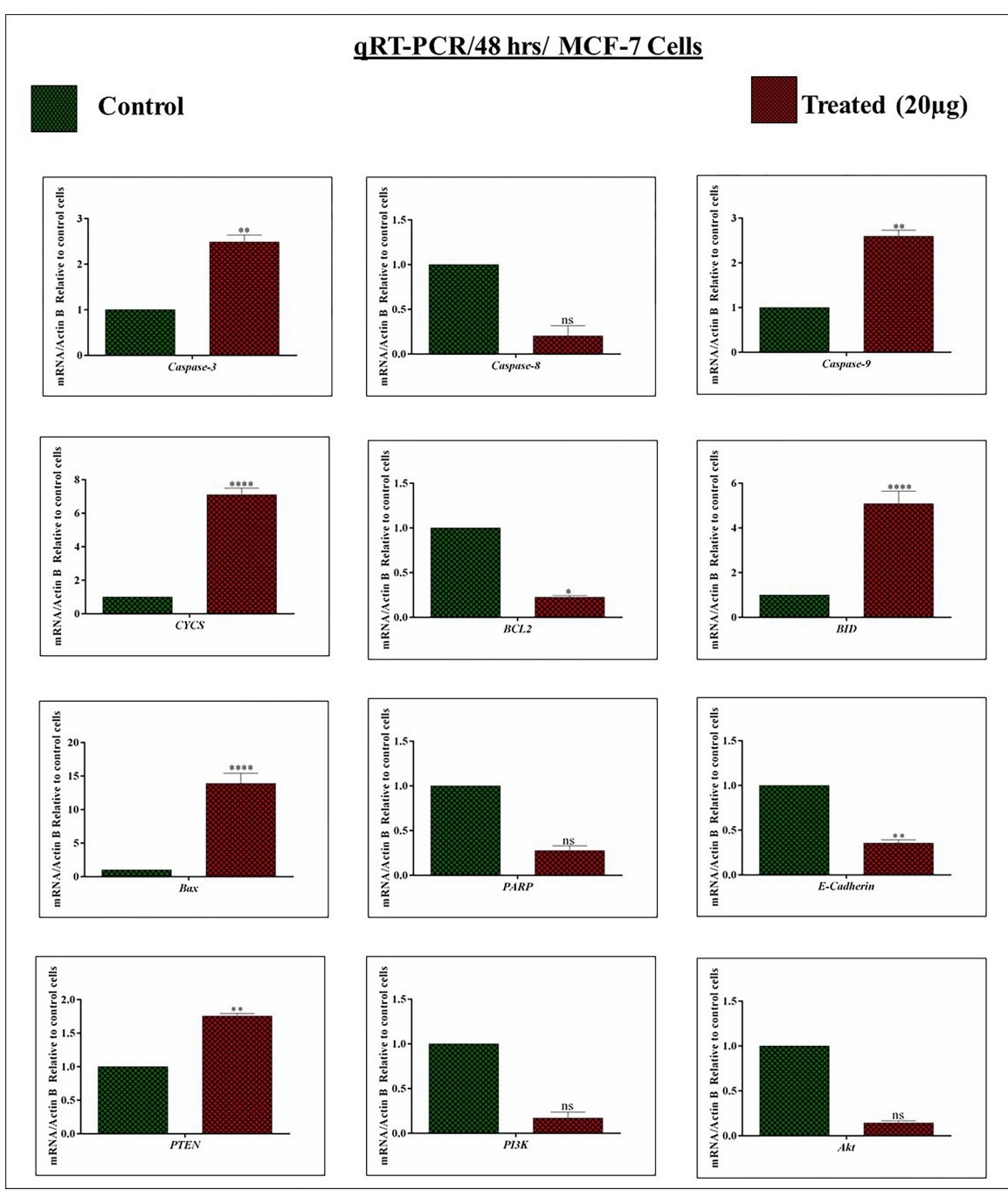

**Fig 6. Impact of *achillea fragrantissima* on apoptosis regulatory genes at 48 hrs (An mRNA account).** The MCF-7 cells were incubated with *achillea fragrantissima* for 48 hrs. The target mRNA such as *Caspase-3*, *Caspase-8*, *Caspase-9*, *Cytochrome c*, *BCL-2*, *BID*, *BAX*, *PARP*, *E-Cadherin*, *PTEN*, *PI3K*, and *Akt* expression was analyzed by qRT-PCR. The *β-Actin* level was utilized as the internal control. All values are represented as mean ± SD (n = 3). Statistically significant at [#] $P<0.0001$, [***] $P<0.001$, [**] $P<0.01$, [*] $P<0.1$ and ns. Statistical significance was performed by Two Way ANOVA/ Dunnett's test.

cleavage more significantly in MCF-7 cells, authenticates the cleaved portion of *PARP*, and helps the impairment of DNA repairing. The ETC mainly relies upon a protein called *Cytochrome c*, and thus expression was analyzed. The *A.fragrantissima* actively stimulated the release of cytoplasmic *CYCS* protein in MCF-7 cells. Brifly, the *A.fragrantissima* upregulates

CYCS protein (0.4 fold) release in the 24 hrs treated cells. Continusoly, it boosts upto 0.5-fold increases while compared with the untreated MCF-7 cells. Interestingly, the *caspase-8* expression was induced in a time-dependent manner translation level. The instability of *caspase-8*-mRNA in the control group leads to a downregulated protein level. That assumes that *caspase-8* mRNA stability is boosted by the *A.fragrantissima* in the treated group. Deeply, under 24 hrs treatment condition induces the expression of *casapase-8* protein 1.1-fold level. Moreover, 2.1-fold is increases under the 48 hrs trated MCF-7 cells. Furthermore, *FADD* protein expression was significantly increased by the *A.fragrantissima* administration as incrased fold levals of 2.6 and 2.9 in 24 and 48 hrs treated MCF-7 cells while compared with untreated cells. Total findings concluded that *PARP* cleavage, *CYCS* releasing, *caspase-8*, and enhanced expression of *FADD* authenticates the *A.fragrantissima* induces apoptosis in breast cancer MCF-7 cells (Fig 7).

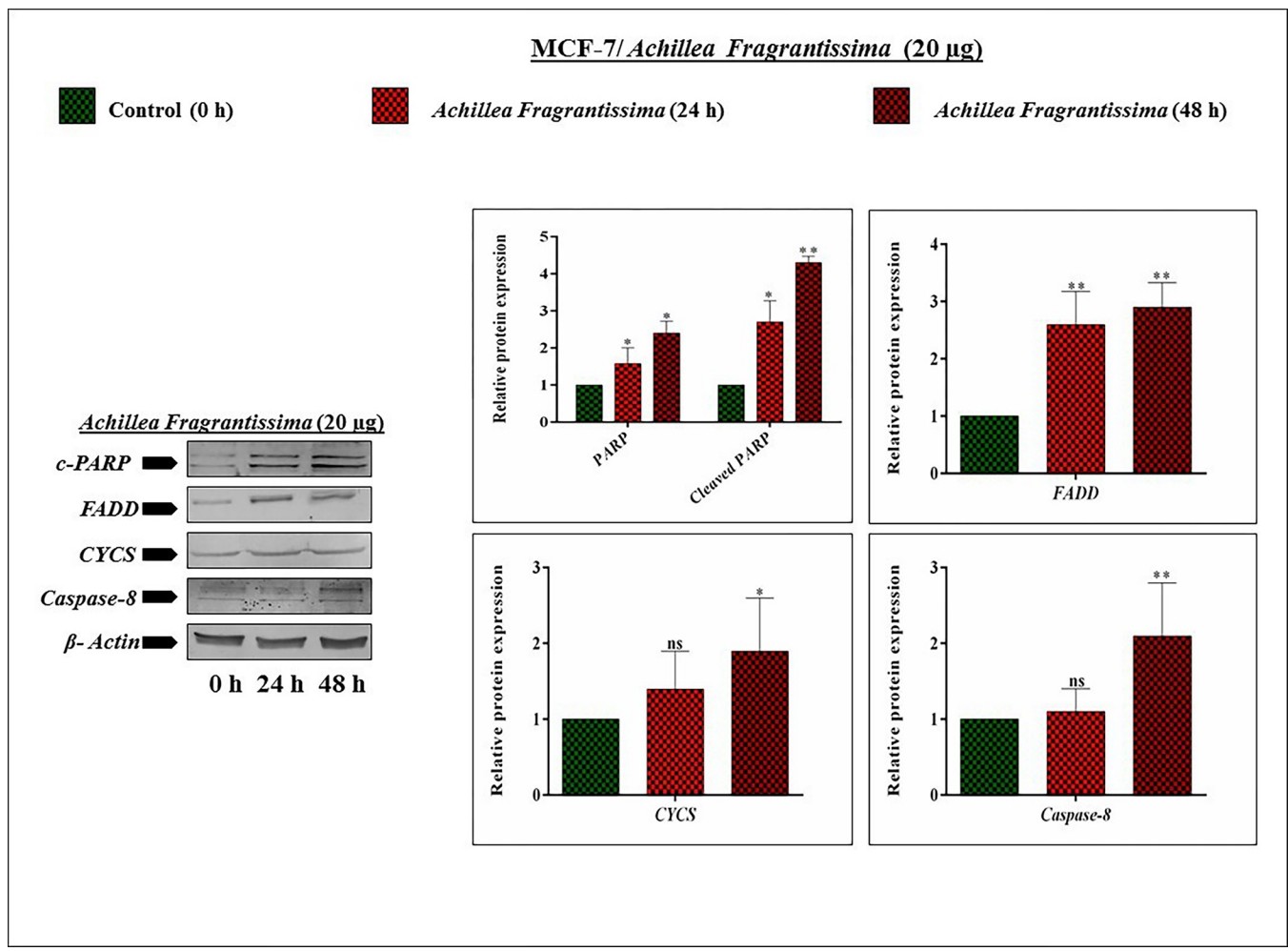

**Fig 7. *PARP*, *FADD*, *and CYSC* proteins are upregulated under *achillea fragrantissima* administration (a protein account) in MCF-7 cells.** The image shows the target protein expressions of *achillea fragrantissima* treated breast cancer MCF-7 cells. The speculated proteins, such as *PARP*, *FADD*, *and CYSC* expression, were tested by the Immuno-blotting method. The *β-Actin* level was used as an internal control of the experiment. Values are represented as mean ± SD (n = 3). Statistically significant at [#] $P<0.0001$, [***] $P<0.001$, [**] $P<0.01$, [*] $P<0.1$ and ns. Statistical significance was performed by Two Way ANOVA/ Dunnett's test.

## 4. Discussion

*Achillea fragrantissima* is a shrub plant found in regions of Arabic countries, and it was used as medicine in ancient days. *Achillea* is a significant genome classified under the *Asteraceae* family that contains over 300 species including *A.fragrantissima* [10, 11]. Aerial parts of this *achillea* were used in folk medicine in Saudi Arabia, followed by the historical hero Achilles during the *A.fragrantissima* Trojan war period [11]. Various studies have been performed, and it illustrates that medicinal properties such as anti-neuroinflammatory [17], anti-angiogenesis [18], anti-metastasis [18], neuroprotection [19], myorelaxant [20], anti-inflammatory [21], anti-diabetic [22, 23], anti-trypanosomal [24], anti-microbial [25, 26], anti-oxidant [27], anti-proliferation [28], pro-apoptotic [28], cytotoxicity [9, 27], and anti-cancer [29–33] effects through the difference solvent extract, prepared oil or phytochemical constituents isolation methods. Despite that, a deep mechanism needs to prove the effect of *A.fragrantissima* on apoptosis function.

In this study, apoptotic properties of *A.fragrantissima* were documented using breast cancer cell MCF-7 as a model. Indeed, we sought the apoptotic regulator expressions upon treatment of *A.fragrantissima*. As a continuation, the preliminary screening authenticates and notifies that increasing concentration of *A.fragrantissima* induces cell cytotoxicity and increases apoptotic cells in breast cancer cells. Consequently, the viability nature of the cell decreases [9, 27]. Also, the generated ROS in breast cancer cells leads the apoptosis, which was well-matched with Redza and Bates [34]. Along with this, ROS upregulated mitochondrial-mediated cell death and ER responsibility [34]. The $IC_{50}$ effects were cross-checked, and the findings were reflected in AO/ EtBr examination. Collectively, our preliminary results suggested that the role of *A.fragrantissima* on stimulation of mitochondrial-mediated cell apoptosis with collapsed cytoplasm.

Next, this study assumes the impact of mitochondria on energy production and sustainability of ATPs upon the treatment condition. Hence, the Rho-123 staining confirmed the permeabilization on the outer membrane of mitochondria and it was well-linked with apoptosis [35]. As a continuation of this exploration, the cleavage of central dogma is considered a fundamental hallmark of apoptosis [36]. Together, this evidence must disintegrate cellular morphology and pave the way for cellular dysfunction. In addition, the increased level of ROS might contribute to the loss in cell migration. This phenomenon was clarified and proved by Yang and his colleagues [37]. They characterized the upregulation of *DLC1* prominently suppressing the *CAV-1* in breast cancer cells. It reflected in wound healing ability loss upon treatment with $CAT/H_2O_2$ [37], and present findings comply with the same.

In addition, an increased amount of extracellular ATP renders the migration property and invasion ability to tumor cells in response to cell death by chemotherapeutic drugs [38]. Conversely, this study reveals that plant-based extraction could arrest the invasion traits of MCF-7 as the response to cellular apoptosis and decreased levels of cellular ATP. Moreover, theATP lacks, resulting in a decreasing amount of new colonies formation. Further, apoptosis relies upon the upregulation of pro-apoptotic genes elevation; regarding this, Bali et al.[39] performed the PCR assay among prostate cancer cells that had been treated with *Achillea teretifolia*. The assay revealed that *BCL2*, *Caspase-3*, and *Bax* genes were significantly expressed, and these findings were more clearly reflected in our present results. Activated pro-apoptotic genes might activate the *Bid* level, and it helps the oligomerization of *Bax* proteins to form the pore in the mitochondria membrane [40]and release the *cytochrome c*. The released *cytochrome c* is connected with *Apaf-1* activation and theATP to form the apoptosome. Caspase cascades are activated and lead to cell apoptosis [41]. Further, the formed apoptosome might cause *PARP* protein cleaving and impair DNA repair. In turn, the cleaved 89 kDa *PARP* fragment activates

the *Apoptosis Induction Factor* (*AIF*) [42]. In addition, apoptosis caused cell adhesion integrity loss, which had been notified through *E- Cadherin* down-regulation [43, 44]. *PTEN* acted as the tumor suppressor role in normal cells and conversely worked and upregulated the life span of mutant cells. The current study explores that *A.fragrantissima* upregulated the tumor suppressor *PTEN* gene and finely downregulated the *PI3K/Akt* mRNAs. These findings are corroborated by the previous documented report [45, 46]. In addition, over-expression of *FADD* instigates apoptosis through the extrinsic pathway and also regulates HCC cell apoptosis [3, 46]. *A.fragrantissima* significantly regulates the apoptosis in MCF-7 cells through intrinsic and extrinsic cell death factors.

The present findings illustrate that the ROS level was boosted by the *A.fragrantissima* ethanolic extraction in MCF-7 cells. The imbalance of free radical ions creates an impact on the tumor suppressor gene and downregulates the cell survival signaling through inhibited anti-apoptotic genes and significantly expressed pro-apoptotic genes. It jointly created the pour in the outer membrane of mitochondria. Altogether, it generates the leaking of *cytochrome c* and it enters to form the apoptosome with activated *caspase-9/Apaf-1*. The caspase cascade helps cleave the mature *PARP* into cleaved *PARP* and affects the DNA repair machinery. The proteolytic enzymes engage with the *FAS* recruitment and activate the *FADD* and signals to the *caspase-8* active form. Finally, cell apoptosis could occur in MCF-7 cells (**Fig 8**). Herein, this study documented that *A.fragrantissima* instigated mitochondrial-mediated apoptosis in

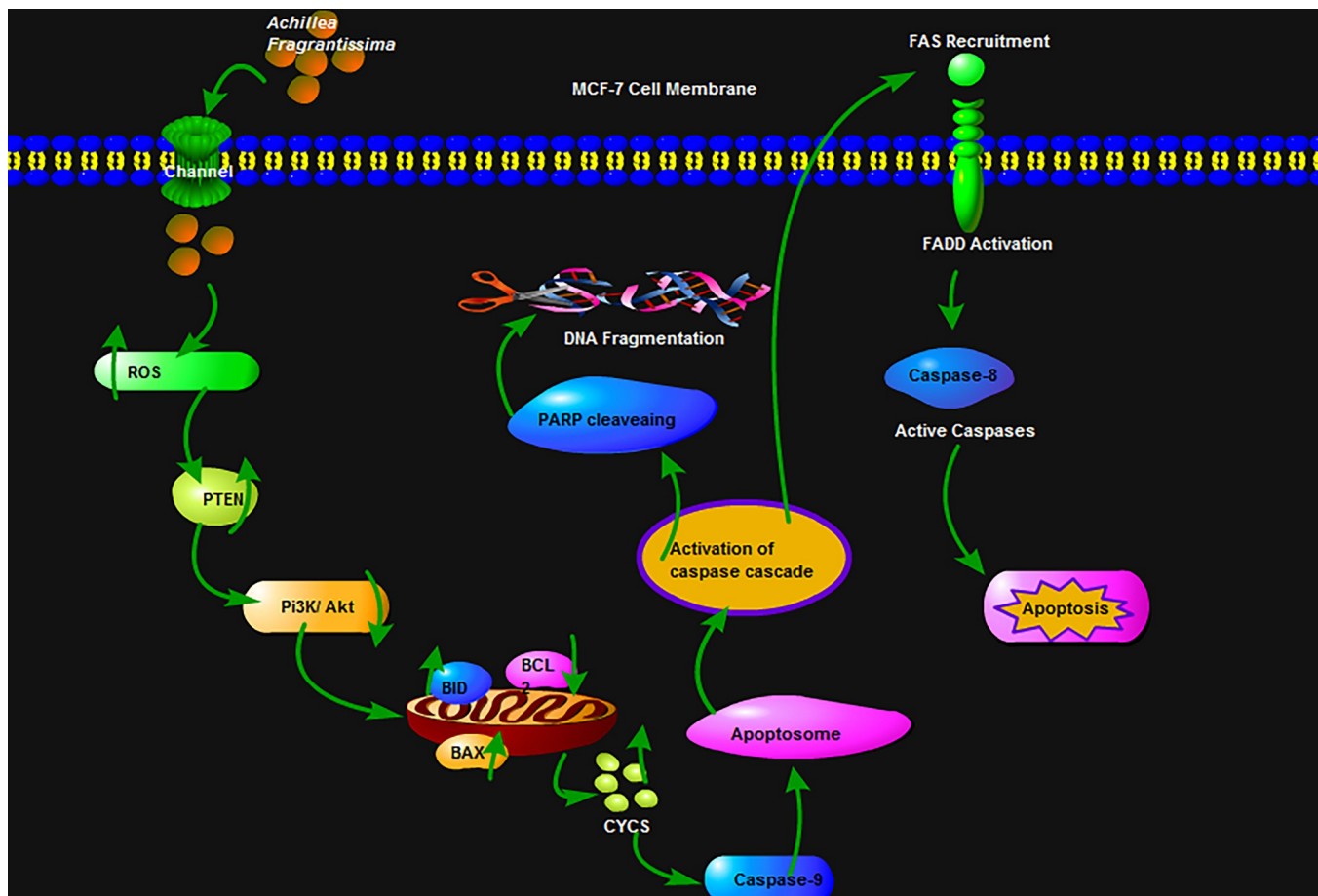

**Fig 8. Schematic representation of the study.** A diagram explains the possible role of *achillea fragrantissima* ethanol extract on MCF-7 cell apoptosis.

breast cancer MCF-7 cells. A recent study has demonstrated the anti-breast cancer activity of the dichloromethane extract, particularly in triple-negative breast cancer (TNBC). This extract exhibited a positive activation of caspase 3/7 activity. Similarly, our study indicated that MCF-7 cells were impacted by the ethanolic extract of *A.fragrantissima*, potentially by influencing the *Pi3K/Akt* signaling cascade [22].

## 5. Conclusion

In conclusion, the findings from this study underscore the cytotoxic potential of the ethanol extract of *A.fragrantissima* against MCF-7 cells, evidencing its ability to induce cell death characterized by the generation of reactive oxygen species (ROS), disruption of mitochondrial potential, and DNA fragmentation. Notably, the extract exhibited promising inhibitory effects on cell migration and invasion, indicative of its potential role in impeding cancer progression. Moreover, *A.fragrantissima* demonstrated a modulatory impact on apoptosis-related genes, notably upregulating key mediators such as *Caspase-8*, *Cytochrome c*, and *FADD*, while inducing significant *PARP* cleavage, a hallmark of apoptotic activity. These collective findings suggest the therapeutic potential of *A.fragrantissima* in breast cancer management. Nonetheless, further investigations, particularly *in vivo* studies, are imperative to elucidate the precise molecular mechanisms underpinning its actions and to validate its efficacy and safety for clinical application.

## Supporting information

**S1 File.**
(DOCX)

## Author Contributions

**Conceptualization:** Abdulrahman Alasmari.

**Investigation:** Abdulrahman Alasmari.

**Methodology:** Abdulrahman Alasmari.

**Writing – original draft:** Abdulrahman Alasmari.

**Writing – review & editing:** Abdulrahman Alasmari.

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
