## [Decision Letter · Decision Letter 0]

6 Feb 2024

PONE-D-23-40260Achillea fragrantissima (Forssk.) Sch.Bip instigates the ROS/FADD/c-PARP expression: An actuation of apoptosis in breast cancer cellPLOS ONE

Dear Dr. ALASMARI,

Thank you for submitting your manuscript to PLOS ONE. After careful consideration, we feel that it has merit but does not fully meet PLOS ONE’s publication criteria as it currently stands. Therefore, we invite you to submit a revised version of the manuscript that addresses the points raised during the review process.

We look forward to receiving your revised manuscript.

Kind regards,

Ajay Kumar, PhD

Academic Editor

PLOS ONE

Journal Requirements:

Reviewers' comments:

Reviewer's Responses to Questions

**Comments to the Author**

1. Is the manuscript technically sound, and do the data support the conclusions?

Reviewer #1: No

Reviewer #2: Yes

Reviewer #3: Partly

2. Has the statistical analysis been performed appropriately and rigorously? 

Reviewer #1: No

Reviewer #2: Yes

Reviewer #3: Yes

3. Have the authors made all data underlying the findings in their manuscript fully available?

Reviewer #1: Yes

Reviewer #2: Yes

Reviewer #3: No

4. Is the manuscript presented in an intelligible fashion and written in standard English?

Reviewer #1: Yes

Reviewer #2: Yes

Reviewer #3: Yes

5. Review Comments to the Author

Reviewer #1: 1. Revise the scientific name of the plant throughout the entire manuscript.

2. Ensure that all gene names are formatted in italics.

3. Authors should reference the supporting papers for the materials and methods section and specify the number of cells used in various in vitro assays.

4. Clearly indicate the specific part of the plant utilized in this study.

5. Rewrite the heading "Determination of Live/Dead assay (AO/EtBr)."

6. Consider replacing "produces" with "induces" in the sentence "The 24-hour experimental time revealed that A. Fragrantissima induces 50% cell death at 20 µg/mL in MCF-7 cells."

7. Based on the results, it appears that the selected plant extract is toxic to normal cells at very low concentrations. If the IC50 value exceeds 100 µg/mL for the normal cell line (HEK293), it can be assumed that the extract is safe for normal cells.

8. Question the need for performing a cell viability experiment if the cell viability has already been assessed through the MTT assay. Consider excluding this experiment from the article.

9. Elaborate on the results section of 3.4, as the material and method for this experiment are missing.

10. To validate cellular integrity, Hoechst or DAPI staining will be used. The colony formation assay only validates the cell migration ability of cancer cells.

11. Authors should provide macroscopic and microscopic images of colonies in the colony formation experiment section.

12. Check the statistical significance in Figure 1d. Authors have marked **** in the image panel.

13. Include a scale bar in Figure 3d.

14. Enhance the language to improve readability and understanding.

15. Avoid using abbreviations in the abstract.

16. In the abstract section, write the name of the plant, Achillea fragrantissima (A. fragrantissima), in the proper format, as well as throughout the entire manuscript.

17. The introductory part should include a brief overview of the current status, overall incidence rates, and epidemiology of breast cancer to make the article more impactful.

18. Address grammatical errors and spelling mistakes throughout the entire article, including line 102, 111, 115, 121, 167, 207, 271.

19. Characterize the ethanolic extract of A. fragrantissima and include information about various bioactive compounds present in the extract.

20. Ensure consistency between the result section 3.1 and Figure 1b, as the normal cells HEK-293 are mentioned as HEK-293T in Figure 1b.

21. Provide the relevance of the present results with previous research to enhance the impact of the article.

22. Specify the criteria for selecting the time intervals in Figures 1d, 3a, and 4.

23. Clearly distinguish between gene and protein for better understanding.

24. Highlight the "toxicity and pharmacokinetic studies" of the ethanolic extract of A. fragrantissima in the article.

25. Improve image quality for better visualization of cell morphology.

26. Ensure that the concentration unit in Figure 1a, b, and c is in μg/mL.

27. In Figure 7, correct the mention of CYSC protein to CYCS in the graphical part.

28. Elaborate on the concluding parts of the article.

Reviewer #2: 1. The manuscript contains several grammatical errors.

2. Inclusion of other breast cancer cell lines alongside MCF-7 will improve the manuscript.

3. In the section 3.1.A, in place of using HEK cells as a control, it is advised to use MCF-10A cells, which is more appropriate control for breast cancer studies.

4. Vehicle control (e.g., ethanol alone) is missing in the ethanol-extracted compound experiment. Further, inclusion of positive control in cell death experiments, such staurosporine or doxycycline induced cell death etc. will be useful.

5. In case of dual staining experiments, apart from merged images, images from the individual channels must be added.

6. Lack of Hoechst 33342 based nucleus staining in the control samples need to be explained.

7. In the mRNA expression analysis, both, PARP and caspase 8 are downregulated. However, in the western-blot based experiments, both are increased. What is the reason behind this discrepancy?

8. In case of cytochrome c release experiment, it is not clear whether the whole cell extract was used or only mitochondrial extract was used.

Reviewer #3: Abdulrahman ALASMARI evaluated the efficacy of Achillea fragrantissima against the breast cancer cell (MCF-7) and claims the possible involvement of ROS/FADD/c-PARP expression for induced apoptotic network genes (Caspase-3, Caspase-8, Caspase-9, Cytochrome c, BCL-2, BID, BAX, PARP, PTEN, PI3K, and Akt).

The manuscript is well designed and scientifically sound, however data analysis should be improved. Discussion part is poorly explained.

I offer the following comments in hope to assist with improvement:

1. Author used ethanol extract of A. Fragrantissima in treating MCF-7 cells. Did they used any appropriate vehicle control for experiment? Include this in manuscript.

2. A. Fragrantissima has shown the decreased cell viability against normal cells - HEK-293 T cells (fig 1). How the result was correlated?

3. I wonder to see the image of DCFH-DA (ROS) and Hoechst-33342 (DNA Fragmentation) stained cells. No cellular expression has been observed in control/MCF-7 cells (Fig 2).

4. The quantitative analysis is poorly defined and need to be elaborated (such as percentage change or modulation-% increase/decrease etc.) for all data.

5. Manuscript title is not conclusive; it could be improved.

6. PLOS authors have the option to publish the peer review history of their article (what does this mean?). If published, this will include your full peer review and any attached files.

Reviewer #1: No

Reviewer #2: **Yes: **Akhilesh Kumar

Reviewer #3: **Yes: **Akhilendra Kumar Maurya

---

## [Author Response · Author response to Decision Letter 0]

17 Apr 2024

Authors Response to Reviewer/Editors Comments

We express our gratitude to the reviewers for their valuable comments and suggestions on this manuscript. We are pleased to present the revised version for potential publication in your esteemed journal.

Reviewer #1:

1. Revise the scientific name of the plant throughout the entire manuscript.

 In accordance with the reviewer's suggestion, we have updated the botanical nomenclature throughout the manuscript.

 2. Ensure that all gene names are formatted in italics.

 We have italicized gene names throughout the entire manuscript and figures as per formatting guidelines.

3. Authors should reference the supporting papers for the materials and methods section and specify the number of cells used in various in vitro assays.

 We have included pertinent references for certain methods, while adhering to the standard protocol for the remaining methods.

4. Clearly indicate the specific part of the plant utilized in this study.

 In the method section, we have specified the particular part of the plant. Thank you.

5. Rewrite the heading "Determination of Live/Dead assay (AO/EtBr).

 We have revised the sub-heading to "Live/Dead Assay (AO/EtBr Staining)".

6. Consider replacing "produces" with "induces" in the sentence "The 24-hour experimental time revealed that A. Fragrantissima induces 50% cell death at 20 µg/mL in MCF-7 cells."

 Following the reviewer's feedback, we have substituted the term "induces". Thank you.

7. Based on the results, it appears that the selected plant extract is toxic to normal cells at very low concentrations. If the IC50 value exceeds 100 µg/mL for the normal cell line (HEK293), it can be assumed that the extract is safe for normal cells.

 If the dosage exceeds 100 µg/mL, it would indeed become toxic to HEK-293 cells.

8. Question the need for performing a cell viability experiment if the cell viability has already been assessed through the MTT assay. Consider excluding this experiment from the article.

 We acknowledge this comment. The temporal dynamics and dose-response relationship of A. Fragrantissima were characterized using the MTT test. We kindly request you to consider including these results in the manuscript.

9. Elaborate on the results section of 3.4, as the material and method for this experiment are missing.

 Section 3.4 has been expanded upon, and the respective method has been included. Thank you.

10. To validate cellular integrity, Hoechst or DAPI staining will be used. The colony formation assay only validates the cell migration ability of cancer cells.

 We concur. Thank you. The Hoechst DNA integrity assay was conducted. Additionally, the colony formation method section has been rectified.

11. Authors should provide macroscopic and microscopic images of colonies in the colony formation experiment section.

 We conducted the colony formation assay and counted the colonies for graphing purposes. The plates were then destroyed after the colonies were counted once. We apologize for any inconvenience caused.

12. Check the statistical significance in Figure 1d. Authors have marked **** in the image panel.

 Upon careful analysis, we have corrected the error.

13. Include a scale bar in Figure 3d.

 The scale bar has been indicated in Figure 3d.

14. Enhance the language to improve readability and understanding.

 The language of the manuscript has been improved.

15. Avoid using abbreviations in the abstract.

 We have corrected the error in the abstract section.

16. In the abstract section, write the name of the plant, Achillea fragrantissima (A. fragrantissima), in the proper format, as well as throughout the entire manuscript.

 The plant name has been rectified throughout the entire manuscript.

17. The introductory part should include a brief overview of the current status, overall incidence rates, and epidemiology of breast cancer to make the article more impactful.

 The introduction section has been enriched with a concise overview of breast cancer.

18. Address grammatical errors and spelling mistakes throughout the entire article, including line 102, 111, 115, 121, 167, 207, 271.

 The grammatical error was corrected.

19. Characterize the ethanolic extract of A. fragrantissima and include information about various bioactive compounds present in the extract.

 In the manuscript, we have detailed the bioactive compounds found in the ethanol-based crude extract of A. Fragrantissima.

20. Ensure consistency between the result section 3.1 and Figure 1b, as the normal cells HEK-293 are mentioned as HEK-293T in Figure 1b.

We corrected the error, and we used HEK-293 cells.

21. Provide the relevance of the present results with previous research to enhance the impact of the article.

 Our current findings were compared with recent findings and it was added in manuscript.

22. Specify the criteria for selecting the time intervals in Figures 1d, 3a, and 4.

 Performing in-vitro studies with various time points enables researchers to thoroughly evaluate temporal dynamics, dose-response relationships, kinetics of cellular processes, and clinical significance of tested treatments. This approach offers valuable insights into their efficacy and mechanism of action.

23. Clearly distinguish between gene and protein for better understanding.

 We differentiated the protein and gene in the manuscript.

24. Highlight the "toxicity and pharmacokinetic studies" of the ethanolic extract of A. Fragrantissima in the article.

 We highlighted the toxicity and pharmacokinetic studies in this article.

25. Improve image quality for better visualization of cell morphology.

 The quality of morphology image was improved.

26. Ensure that the concentration unit in Figure 1a, b, and c is in μg/mL.

The concentration unit was corrected.

27. In Figure 7, correct the mention of CYSC protein to CYCS in the graphical part.

 We corrected “CYSC into CYCS”. Thank you.

28. Elaborate on the concluding parts of the article.

 Conclusion section was rewrite with necessary words. 

Reviewer #2:

1. The manuscript contains several grammatical errors.

 As per the reviewer suggestions, the several grammatical error was corrected. 

2. Inclusion of other breast cancer cell lines alongside MCF-7 will improve the manuscript.

 Thank you for your comment. We have concentrated on investigating the impact of A. Fragrantissima on MCF-7 cells, which serve as a prominent cell line model for breast cancer studies.

3. In the section 3.1.A, in place of using HEK cells as a control, it is advised to use MCF-10A cells, which is more appropriate control for breast cancer studies.

 Due to the availability of a widely used normal cell line, HEK-293 was utilized. Additionally, several important articles have recommended the use of HEK-293 as normal cells for breast cancer studies. Thank you.

4. Vehicle control (e.g., ethanol alone) is missing in the ethanol-extracted compound experiment. Further, inclusion of positive control in cell death experiments, such staurosporine or doxycycline induced cell death etc. will be useful.

 Thank you for your comment. From a technical standpoint, we employed a lyophilization step to remove ethanol solvent from the crude extract. Consequently, ethanol could not influence the growth of the cell line or induce cell death in MCF-7 cells. This is why we opted not to conduct a vehicle control test. 

We did not utilize a positive control for screening apoptosis. The rationale behind this decision is that we did not compare our extracted compound with other anti-cancer agents. Moreover, the plant extract unequivocally induces cell death. Considering these factors, we omitted the positive control from this study. Thank you.

5. In case of dual staining experiments, apart from merged images, images from the individual channels must be added.

 We performed the dual staining assay as a merged manner. Excuse this inconvenience. 

6. Lack of Hoechst 33342 based nucleus staining in the control samples need to be explained.

 The action of Hoechst staining on control cells had explained.

7. In the mRNA expression analysis, both, PARP and caspase 8 are downregulated. However, in the western-blot based experiments, both are increased. What is the reason behind this discrepancy?

 We are also surprised by these results. The explanation lies in the fact that control cells do not undergo apoptosis. Consequently, during the protein translation stage, PARP and Caspase-8 proteins undergo normal cellular degradation and cleavage. However, in the case of treated cells, these proteins remain stable throughout the treatment and induce apoptosis. Thank you.

8. In case of cytochrome c release experiment, it is not clear whether the whole cell extract was used or only mitochondrial extract was used.

 We are using whole cell lysate for western blot analysis. Now, we mentioned it method section and thank you.

Reviewer #3:

Abdulrahman ALASMARI evaluated the efficacy of Achillea fragrantissima against the breast cancer cell (MCF-7) and claims the possible involvement of ROS/FADD/c-PARP expression for induced apoptotic network genes (Caspase-3, Caspase-8, Caspase-9, Cytochrome c, BCL-2, BID, BAX, PARP, PTEN, PI3K, and Akt). The manuscript is well designed and scientifically sound, however data analysis should be improved. Discussion part is poorly explained. I offer the following comments in hope to assist with improvement:

1. Author used ethanol extract of A. Fragrantissima in treating MCF-7 cells. Did they used any appropriate vehicle control for experiment? Include this in manuscript.

 Thank you for your feedback. As a technical measure, we implemented a lyophilization step to remove ethanol solvent from the crude extract. This ensured that ethanol did not influence the growth of MCF-7 cells or trigger cell death. Hence, we opted not to conduct a vehicle control test.

Additionally, we did not incorporate a positive control for apoptosis screening. This decision stems from the fact that we did not compare our extracted compound with other anti-cancer agents. Moreover, the plant extract unequivocally induces cell death. Considering these factors, we omitted the positive control from this study. Thank you.

2. A. Fragrantissima has shown the decreased cell viability against normal cells - HEK-293 T cells (fig 1). How the result was correlated?

 The plant extract also triggers cell death in HEK-293 cells, albeit at a higher concentration compared to cancerous cell lines. Thank you.

3. I wonder to see the image of DCFH-DA (ROS) and Hoechst-33342 (DNA Fragmentation) stained cells. No cellular expression has been observed in control/MCF-7 cells (Fig 2).

 The images are now corrected. Thank you.

4. The quantitative analysis is poorly defined and need to be elaborated (such as percentage change or modulation-% increase/decrease etc.) for all data.

 Agreed and the western blot result was elaborated.

5. Manuscript title is not conclusive; it could be improved.

 As per reviewer suggestions the title was altered in the revised manuscript. 

*****

---

## [Decision Letter · Decision Letter 1]

7 May 2024

Achillea fragrantissima (Forssk.) Sch.Bip instigates the ROS/FADD/c-PARP expression: An actuation of apoptosis in breast cancer cell

PONE-D-23-40260R1

Dear Dr. ALASMARI,

We’re pleased to inform you that your manuscript has been judged scientifically suitable for publication and will be formally accepted for publication once it meets all outstanding technical requirements.

Kind regards,

Ajay Kumar, PhD

Academic Editor

PLOS ONE

Additional Editor Comments (optional):

The authors have adequately addressed all the comments asked by the reviewers and have accordingly revised the manuscript. Hence, the manuscript can be accepted for publication in its present form.

Reviewers' comments:

Reviewer's Responses to Questions

**Comments to the Author**

1. If the authors have adequately addressed your comments raised in a previous round of review and you feel that this manuscript is now acceptable for publication, you may indicate that here to bypass the “Comments to the Author” section, enter your conflict of interest statement in the “Confidential to Editor” section, and submit your "Accept" recommendation.

Reviewer #1: All comments have been addressed

Reviewer #2: All comments have been addressed

Reviewer #3: All comments have been addressed

2. Is the manuscript technically sound, and do the data support the conclusions?

Reviewer #1: Partly

Reviewer #2: Yes

Reviewer #3: Yes

3. Has the statistical analysis been performed appropriately and rigorously? 

Reviewer #1: Yes

Reviewer #2: Yes

Reviewer #3: Yes

4. Have the authors made all data underlying the findings in their manuscript fully available?

Reviewer #1: Yes

Reviewer #2: Yes

Reviewer #3: Yes

5. Is the manuscript presented in an intelligible fashion and written in standard English?

Reviewer #1: Yes

Reviewer #2: Yes

Reviewer #3: Yes

6. Review Comments to the Author

Reviewer #1: The manuscript has undergone a comprehensive review process. All the comments, suggestions, are addressed. Recommended for publication.

Reviewer #2: (No Response)

Reviewer #3: Authors incorporated all the suggestions and concern; and addressed properly in manuscript for its betterment. I recommend my acceptance for publication.

7. PLOS authors have the option to publish the peer review history of their article (what does this mean?). If published, this will include your full peer review and any attached files.

Reviewer #1: No

Reviewer #2: **Yes: **Akhilesh Kumar

Reviewer #3: **Yes: **Akhilendra Kumar Maurya

---

## [Editor Report · Acceptance letter]

18 May 2024

PONE-D-23-40260R1 

PLOS ONE

Dear Dr. Alasmari, 

I'm pleased to inform you that your manuscript has been deemed suitable for publication in PLOS ONE. Congratulations! Your manuscript is now being handed over to our production team.

Kind regards, 

on behalf of

Dr. Ajay Kumar 

Academic Editor

PLOS ONE